# Tuning the Doping Ratio and Phase Transition Temperature of VO_2_ Thin Film by Dual-Target Co-Sputtering

**DOI:** 10.3390/nano9060834

**Published:** 2019-06-01

**Authors:** Xu Chen, Mingfei Wu, Xingxing Liu, Ding Wang, Feng Liu, Yuwei Chen, Fei Yi, Wanxia Huang, Shaowei Wang

**Affiliations:** 1School of Materials Science and Engineering, University of Shanghai for Science and Technology, Shanghai 200093, China; cxzhixin0211@163.com (X.C.); wangding@usst.edu.cn (D.W.); 2State Key Laboratory of Infrared Physics, Shanghai Institute of Technical Physics, Chinese Academy of Sciences, Shanghai 200083, China; 1000398609@smail.shnu.edu.cn (M.W.); xxliu@mail.sitp.ac.cn (X.L.); 3Department of Physics, Shanghai Normal University, Shanghai 200234, China; fliu@shnu.edu.cn; 4Department of Remote Sensing and Photogrammetry, Finnish Geospatial Research Institute, Geodeetinrinne 2, 02431 Kirkkonummi, Finland; 5School of Optical and Electronic Information, Huazhong University of Science and Technology, Wuhan 430074, China; feiyi@mail.hust.edu.cn; 6College of Material Science and Engineering, Sichuan University, Sichuan 610064, China; huangwanxia@scu.edu.cn

**Keywords:** VO_2_, thin films, doping, co-sputtering, phase transition

## Abstract

A new simple way for tuning the phase transition temperature (PTT) of VO_2_ thin films has been proposed to solve the problem of changing the doping ratio by using the dual-target co-sputtering method. A series of samples with W doping ratios of 0%, 0.5%, 1%, 1.5% and 2% have been fabricated by sputtering V films with the power of pure and 2% W-doped V targets from 500 W: 0 W, 500 W: 250 W, 500 W: 500 W, 250 W: 500 W to 0 W: 500 W respectively and then annealed in an oxygen atmosphere to form VO_2_. The XRD results of both pure and W-doped VO_2_ samples reveal that VO_2_ forms and is the main component after annealing. The PTT can be tuned by controlling the sputtering power ratio of the pure and doped targets. It can be tuned easily from 64.3 °C to 36.5 °C by using the pure and 2% W-doped targets for demonstration, with W doping ratios from 0% to 2%. It is also valid for other doping elements and is a promising approach for the large-scale production of sputtering.

## 1. Introduction

Vanadium dioxide (VO_2_) has been extensively studied in the last few decades because it has a metal-insulator transition (MIT) at about 68 °C [1], which results in changes in both its electrical and optical properties. An MIT near room temperature (RT) makes a good candidate material for a wide range of applications, such as uncooled infrared (IR) detector [2], optical switching [3] and ‘smart window’ which can be used to control the room temperature intelligently for energy-saving [4]. 

However, many of the above applications need VO_2_ to work at near RT to have high efficiency and good performance. Therefore it is important to lower the phase transition temperature (PTT) of VO_2_, and many efforts have been made to do so. The PTT can be influenced and tuned by grain size [5,6], stress [7], nanostructure [8,9], and so forth. However their controllability and repeatability is low. Recently it was reported that the PTT of VO_2−x_ thin films on sapphire can be tuned by the O_2_ pressure in the deposition process by pulsed laser deposition [10] and in the post-annealing process by magnetron sputtering [11]. However, the films are deposited at a temperature of 400 °C [5]. It is difficult to heat the whole chamber of an industrial magnetron sputtering coating system and keep its vacuum. They are valid for sapphire substrates, but have not been demonstrated on amorphous glass substrates. Liu et al. reported a new approach for tuning PTT of VO_2_ thin films simply by annealing in atmosphere, which works on both crystalline and amorphous substrates [12]. The PTT can be easily tuned from the intrinsic temperature to 40.2 °C by tuning the annealing oxygen pressure with an RT sputtering. However it still cannot be as low as the RT of 25 °C. 

Doping is another effective method for tuning the PTT of VO_2_. It can tune the PTT in a wide range. Doping elements such as W [13], Mo [14], Ti [15], and Ni [16] were studied extensively. The PTT can be reduced to as low as 20 °C by doped W [17,18,19]. However, it is difficult to change the doping quantity arbitrarily and continuously, especially for the sputtering method which is one of the best industrialized fabrication techniques. For sputtering, changing the doping element or ratio needs a new large-area target. 

In this paper, we present a new approach to fabricate the doped VO_2_ thin films by dual-target co-sputtering without heating, which can tune the doping ratio and resultant PTT according to requirements. It is demonstrated with W-doped VO_2_ thin films and is valid for other doping elements. 

## 2. Experimental Procedure

VO_2_ thin films can be fabricated in two steps using the dual-target co-sputtering method. Firstly, V metal thin films are coated by a homemade large area magnetron sputtering system without heating. The photo and detailed information of the magnetron system have already been reported in other places [12]. Then they are oxidation annealed by controlling the oxygen pressure. The films can be doped by dual-target co-sputtering as shown in Figure 1. One target is pure V and the other is the doped one. The doping element or mixed elements can be pre-made in the doped target with maximum doping ratio. The pure or maximum doped VO_2_ thin film can be fabricated by sputtering the single target of pure or maximally doped V target, respectively. VO_2_ thin film with other doping ratios can be fabricated by sputtering both of the targets simultaneously. The doping ratio can be tuned by controlling the sputtering power of each target, and do not need any new targets with different doping ratios. One can obtain VO_2_ thin film doped with another element or mixed elements simply by changing the target with the desired elements. Such a method can even be used to fabricate VO_2_ thin films with different elements’ co-doping ratios. Thus it is a highly flexible deposition method. 

In this paper, W-doped VO_2_ thin films with different doping ratios were fabricated by dual-target co-sputtering as examples. The planar metal targets with sizes of 300 mm × 140 mm were used as the cathode. W-doped V film was deposited by dual-target co-sputtering method without heating. A pure V target (with purity of 99.99%) and a W-doped target (with *V*: W atom ratio of 98:2) was sputtered at the same time. Different doping concentrations can be achieved by adjusting the sputtering power of each target. Power control mode is selected in the sputtering process. The cleaned substrate was placed in the chamber of the magnetron sputtering coating system and vacuumed to lower than 9.9 × 10^−4^ Pa. The substrate can be amorphous glass or crystalline sapphire which is amorphous K9 glass in this work. The maximum size of samples was 200 mm square which is only confined by the size of the deposition chamber. Argon gas with a flow rate of 50 sccm was injected into the coating chamber as working gas and kept the working pressure at about 2.1 × 10^−2^ Pa. The sputtering power was set to 500 W, which was increased from 0 W to 500 W at a rate of 100 W per minute. The cathode voltage was 322 V. The powers of V target and W-doped V target were set as 500 W: 0 W, 500 W: 250 W, 500 W: 500 W, 250 W: 500 W, 0 W: 500 W, respectively. The targets were sputtered for 30 min before sputtering onto the substrate. The deposition temperature was 25 °C without need to heat the substrate. The substrate moved backward and forward underneath the targets at a constant speed of 1 m per minute, to ensure homogeneity when sputtering. The W-doped V films with different doping ratios were prepared by the above process. The sputtering power deviation is less than 340 V ± 5 V and corresponds to a maximum error of 1.5%. Since the sputtering rate of the two targets is proportional to the power, the deposition ratio of V and W-doped V is calculated to be 2:1, 2:2 and 1:2 for power ratios of 500 W: 250 W, 500 W: 500 W and 250 W: 500 W, respectively. The power ratios of 500 W: 0 W and 0 W: 500 W correspond to samples of pure and 2% W-doped V thin films, respectively. Therefore, the amount of W-doping in the samples prepared at the corresponding power ratio is 0%, 0.5%, 1%, 1.5%, 2%, respectively, since the V: W atom ratio of W-doped V target is 98:2. Five ratios of W-doped V films were fabricated for demonstration by tuning the power of the two targets. 

The resultant films were annealed in a 250 mm diameter tubular vacuum annealing furnace. It was evacuated down to 1.5 Pa by a mechanical pump to remove the air in it. The above films were treated for heating, annealing and cooling stages to form VO_2_ films. The annealing temperature and time were 530 °C and 30 min, respectively. Oxygen (99.999%) was used only in the annealing stage with a flow rate of 20 sccm to ensure the oxygen pressure stayed at 18 Pa. The pressure of oxygen in the furnace was precisely controlled by a mass flow controller and monitored by a vacuum gauge. 

Scanning electron microscopy (SEM, FEI SIRION 200, Hillsboro, OR, USA) was used to observe the surface micrograph of the films. The crystalline structures of the films were characterized by using X-ray diffraction (XRD, BRUKER AXS series D8 ADVANCE, Karlsruhe, Germany) with Cu-Kα radiation, 40 kV/40 mA, 0.02° 2θ step size and scan time of 0.2 s per step in the range of 10°–70° 2θ. Near-infrared transmission spectra of V thin films with different doping ratios were characterized by a NIR2500 infrared fiber optic spectrometer from Ideaoptics Inc. (Shanghai, China) at different temperatures. 

## 3. Results and Discussion

In order to keep the comparability of the fabricated films and increase the efficiency, a combinatorial deposition method [20,21,22,23] has been introduced to fabricate a series of V thin films with different thicknesses. The mask plates with different windows help the selective deposition on a substrate. 2*^n^*−1 thicknesses can be realized by only *n* times of sputtering. The total thickness of film sputtered with 5 cycles (a single cycle is defined as once the substrate moves backward and forward underneath the targets one time) was 45 nm obtained by the cross-section of SEM, and the thickness of each sputtering cycle was deduced from the result. Here V thin films were coated three times with thicknesses of 9 nm, 18 nm and 36 nm respectively for the combinatorial deposition technique. Then 7 V film samples with thicknesses of 9 nm, 18 nm, 27 nm, 36 nm, 45 nm, 54 nm, and 63 nm were obtained. The uniformity of VO_2_ thin film over the maximum size of 200 mm × 200 mm is good which can be seen from the photo presented in reference [24]. 

The micro-structure of pure and 2% W-doped V films was observed by SEM before and after annealing for comparison. The SEM images are presented in Figure 2. Nano-sized grains are uniformly distributed over the entire substrate before annealing. It is almost the same for both the pure and 2% W-doped V films. The distinctive morphological changes have been observed after the annealing process. Many grains exist on the film of pure ones after being annealed in oxygen. Most of them are larger than 100 nm. For the 2% W-doped one, a similar amount of grains was present with sizes of ten nanometers, which are much smaller than the pure one. The difference might be ascribed to the restraint of grain growth by the doping element of W [12]. It will definitely influence the optical and electrical properties of the vanadium oxide thin films. The XRD curves of pure and W-doped VO_2_ samples are presented in Figure 3. Both the pure and W-doped VO_2_ samples have obvious VO_2_ and V_2_O_5_ diffraction peaks. There is also one weak V_6_O_11_ diffraction peak present in the W-doped samples. No other vanadium oxide peaks have been observed. This is because the W atoms replace some V atoms in the VO_2_ polycrystalline film in the form of W^6+^ and form a V_1-x_W_x_O_2_ solid solution [25]. The diffraction peaks of XRD results agree well with the standard cards. The largest deviation of 1.4% comes from the first peak of V_2_O_5_ at 12.404 degrees from the standard card. The slight deviation of V_2_O_5_ peak at 29.14 degrees may be ascribed to the difference between our fabrication method and the method from the standard card. Different oxides and orientations observed should be ascribed to the fabrication of thin films on amorphous substrates of K9 glass. 

The modulation ability of VO_2_ thin films on infrared transmission is critical for its application as intelligent energy-saving glass. It is studied by measuring the transmission spectra before and after phase transition. The results of 0%, 1% and 2% W-doped VO_2_ films are shown in Figure 4, corresponding to VO_2_, V_0.99_W_0.01_O_2_ and V_0.98_W_0.02_O_2_ respectively. For the test error, the signal-to-noise ratio (SNR) of the test spectrum is larger than 10,000 for all the spectral measurements and is good enough. When temperature is lower than the PTT, the films are in the insulation phase and have high infrared transmissions of 49.8%, 57.9% and 52.8% at the wavelength of 2400 nm. They will drop to 18.8%, 14.9% and 10.8% when the temperature is higher than the PTT, because they will change to metal phase with high carrier concentration. Therefore, the infrared transmission changes of pure VO_2_ thin film, 1% and 2% W-doped VO_2_ films are 31%, 43% and 42%, respectively, which reveals a good infrared radiation tunability for energy-saving. The results indicate that the infrared transmission change before and after PTT for W-doped VO_2_ films is larger than that of pure VO_2_ film. The modulation ability of VO_2_ thin films on infrared transmission has been enhanced by doping the W element. There are some other results reported with W-doped VO_2_ thin films. One-dimensional W-doped VO_2_ (M) solid solutions were synthesized under hydrothermal condition and a transmission change of up to 25% between room temperature and 100 °C was observed at 4000 cm^−1^ for the 0.5% W-doped one [26]. Piccirillo et al. used the aerosol assisted chemical vapor deposition method to deposit W-doped vanadium dioxide thin films and observed noticeable changes of transmittance up to about 30% at 2500 nm [27]. 

In the visible range, the transmission spectra of the pure and W-doped VO_2_ films stay almost the same when wavelength is shorter than 440 nm, and have a small difference between 440 and 820 nm before and after the PTT. Both the room temperature and high temperature transmission spectra of doped VO_2_ films in the visible spectrum are a little lower than those of pure VO_2_ film. 

The hysteresis loop of VO_2_ is an important way to study its phase transition process. Figure 5 shows an optical thermal hysteresis loop of the V_0.99_W_0.01_O_2_ film at 2000 nm, which is a result of the phase transition of VO_2_. The transmission decreases with the increase of temperature when heating, and increases with the decrease of temperature when cooling. The PTT (*T_t_*) of the VO_2_ thermochromic films is defined to be *T_t_* = (*T_t,h_* + *T_t,c_*)/2, where *T_t,h_* and *T_t,c_* are the PTTs when heating and cooling, respectively [12]. The *T_t,h_*, *T_t,c_* and corresponding *T_t_* are 48 °C, 38 °C and 43 °C, respectively. 

PTT is a very important parameter of VO_2_ thin film. As mentioned in the experimental section, it can be tuned by changing the doping ratio which is realized by controlling the sputtering power of each target. It is 500 W: 0 W, 500 W: 250 W, 500 W: 500 W, 250 W: 500 W and 0 W: 500 W of pure: 2% W-doped V targets, corresponding to doping ratios of 0%, 0.5%, 1%, 1.5% and 2% respectively for demonstration. The optical thermal hysteresis loops of VO_2_ films at 2000 nm with different W doping ratios are shown in Figure 6a. It can be seen that the optical thermal hysteresis loops move to lower temperatures with an increase of doping ratio. According to the definition of PTT mentioned above, it is 64.3 °C for the pure VO_2_ film without doping and reduces with the increase of doping ratio. The relationship between the PTT and doping ratio of W is shown in Figure 6b. The results come from three repeated experiments and the deviation of PTT is no more than 3%. When the W doping ratio is 0.5%, the PTT decreases 9.5 °C to 54.8 °C. It is lowered to 46.2 °C when the doping ratio increases to 1%. The PTT is further reduced to 41.3 °C when the W content increases to 1.5%. The PTT is down to 36.5 °C for 2% doping ratio, which can be further lowered by using a target with a larger W doping ratio. The transition temperature is reduced to 55 °C and 35 °C with VO_2_-W-1% and VO_2_-W-2% respectively, which is reported by Liu et al. using the aqueous sol-gel method [13] and similar to ours for the 2% doping one. Zhang et al. reported a decrease to 55.3 °C and 45.7 °C respectively with VO_2_-W-1% and VO_2_-W-2% synthesized by a facile hydrothermal approach and subsequent calcination [14], which is higher than ours for the 2% doping one. Li et al. used ion beam enhanced deposition and post-annealing techniques to prepare 3% W-doped VO_2_ on SiO_2_ substrates which reduced the PTT of VO_2_ thin films to 28 °C [19]. The PTT should decrease further if we fabricate W-doped VO_2_ thin film with a higher doping ratio. According to the existing theory, the PTT of VO_2_ film is reduced by W doping, which may be due to the substitution of W^6+^ to V^4+^ in *d* orbital. The introduction of excess electrons reduces the band gap of the *d* orbit and reduces the PTT [19]. 

The results show that the PTT of VO_2_ thin films can be easily tuned by controlling the sputtering power ratio of the pure and doped targets. The maximum doping ratio depends only on the doping ratio of the doped target, which can be adjusted flexibly in a wide range. It is especially good for large-scale production to reduce the cost. In fact, it is not only feasible for the fabrication of W-doped VO_2_ thin films, but also useful for doping other elements such as Mo, Ti, Ni, Cr, Ta or even their mixtures, since the targets are fabricated by mixing different doping particles with V particles and sintered together. 

## 4. Conclusions

In this paper, a dual-target co-sputtering method has been proposed to fabricate W-doped VO_2_ thin films with different doping ratios. It is demonstrated by using a pure V target and a 2% W-doped target to fabricate VO_2_ thin films with doping ratios from 0% to 2%. It is 64.3 °C for the pure VO_2_ film without doping and reduces with the increase of doping ratio. When the W doping ratio is 0.5%, the PTT decreases 9.5 °C to 54.8 °C. It decreases to 46.2 °C when the doping ratio increases to 1%. The PTT is further reduced to 41.3 °C when the W content increases to 1.5% and down to 36.5 °C for a 2% doping ratio. The corresponding PTT of VO_2_ thin films can be readily adjusted from 64.3 °C to 36.5 °C. Such a method can also be used for doping with other elements or mixed doping. It is a prospective approach for arbitrarily doping content and ratio control and large-scale production of sputtering to cut the cost. 

## Figures and Tables

**Figure 1 nanomaterials-09-00834-f001:**
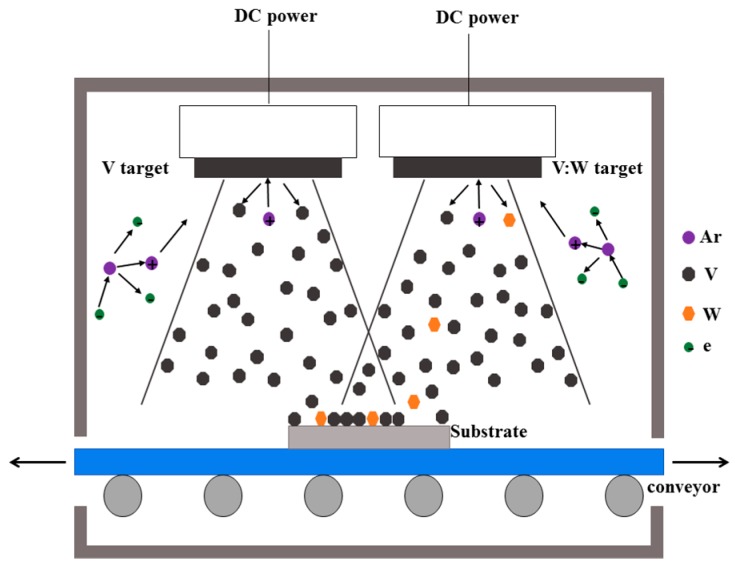
Schematic diagram of fabricating W-doped V thin films with the dual-target co-sputtering method. A pure and a W-doped V targets were used in the co-sputtering procedure. The resultant co-sputtered thin film was oxidation annealed by controlling the oxygen pressure to form the W-doped VO_2_ thin film.

**Figure 2 nanomaterials-09-00834-f002:**
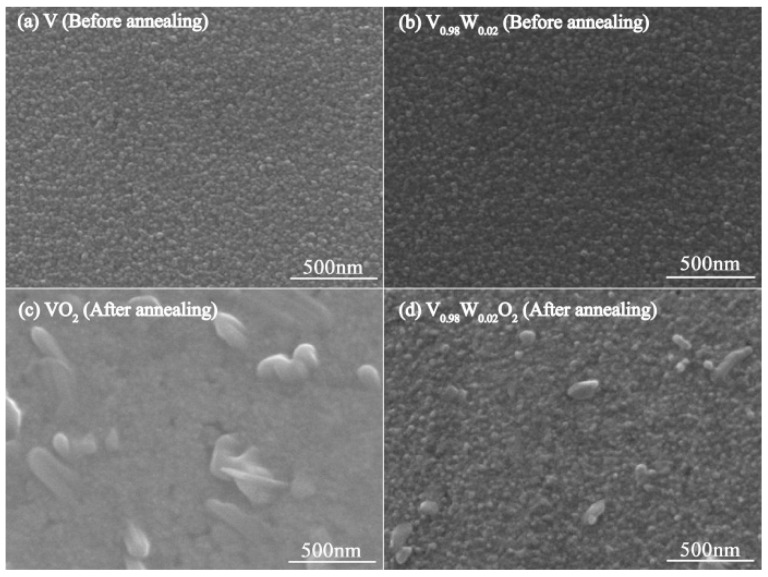
SEM surface micrographs of pure and 2% W-doped V films before and after annealing. (**a**) Pure V film before annealing. (**b**) 2% W-doped V film before annealing. (**c**) Pure V film after annealing. (**d**) 2% W-doped V film after annealing. The films were sputtered with the power of 500 W: 0 W and 0 W: 500 W for the pure and 2% W-doped V targets without heating, respectively. They were annealed at 530 °C for 30 min with oxygen pressure of 18 Pa.

**Figure 3 nanomaterials-09-00834-f003:**
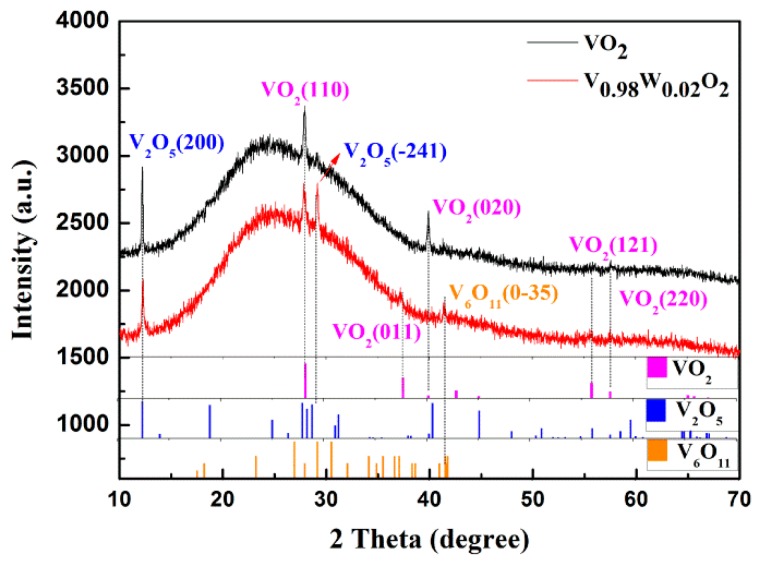
XRD curves of pure and W-doped VO_2_ films fabricated on amorphous K9 glass substrates. The films were sputtered with the power of 500 W: 0 W and 0 W: 500 W for the pure and 2% W-doped V targets without heating, respectively. They were annealed at 530 °C for 30 min with oxygen pressure of 18 Pa.

**Figure 4 nanomaterials-09-00834-f004:**
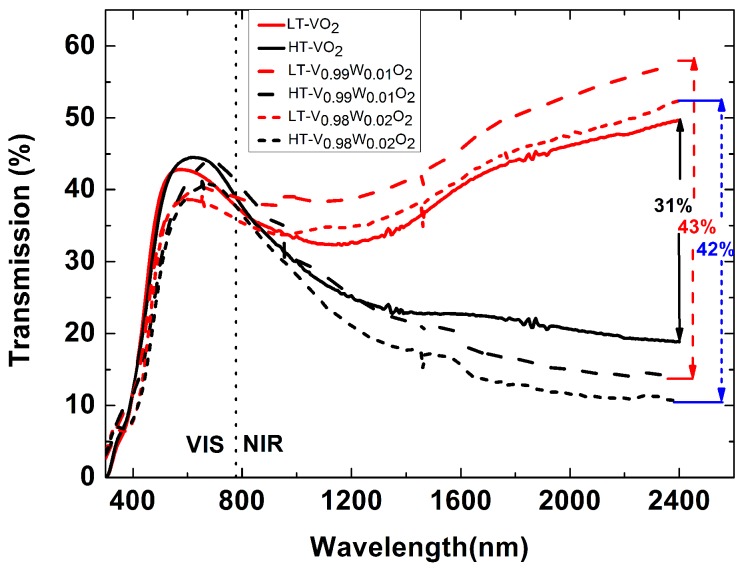
Visible and near-infrared transmission spectra of pure VO_2_, V_0.99_W_0.01_O_2_ and V_0.98_W_0.02_O_2_ films before (10 °C) and after (100 °C) PTT, respectively. The films were annealed at 530 °C for 30 min with oxygen pressure of 18 Pa.

**Figure 5 nanomaterials-09-00834-f005:**
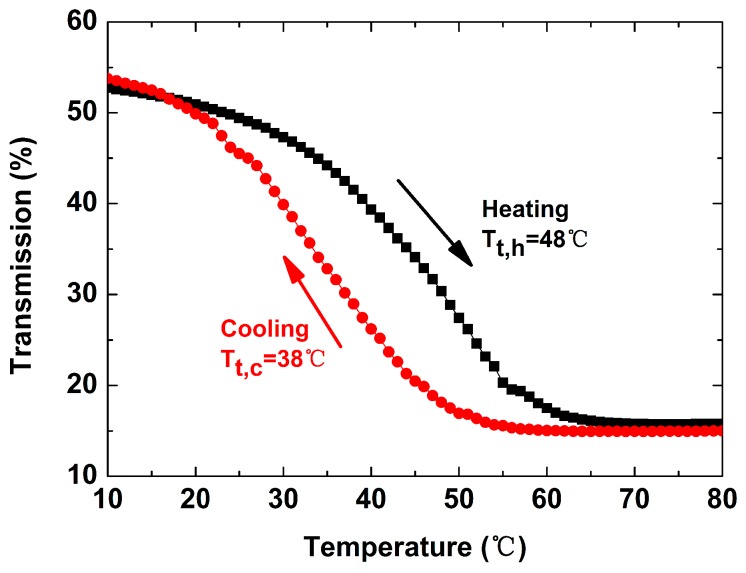
Thermal hysteresis loop of V_0.99_W_0.01_O_2_ film at 2000 nm. The black curve indicates that the transition temperature is 48 °C when the sample is heated, and the red curve indicates that the transition temperature is 38 °C when the sample is cooled.

**Figure 6 nanomaterials-09-00834-f006:**
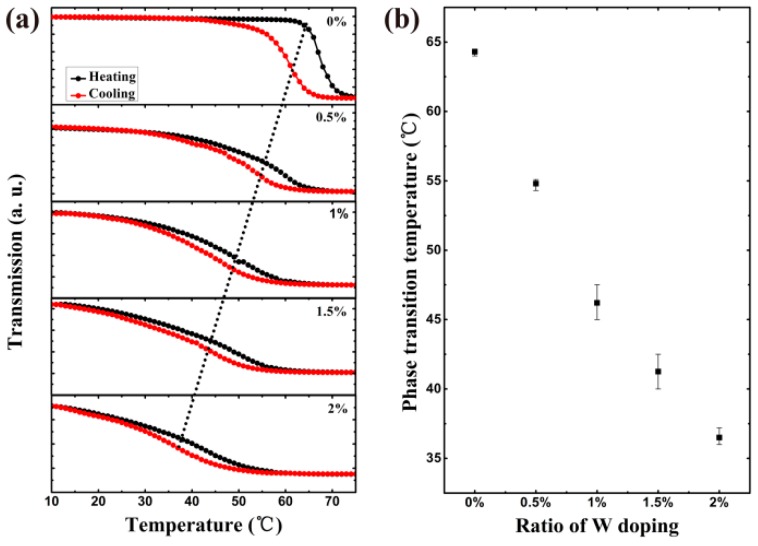
(**a**) Optical thermal hysteresis loop of VO_2_ thin films with different W doping ratios. (**b**) Relationship between the PTT and doping ratio of W. The results come from three repeated experiments and the deviation of PTT is no more than 3%.

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
