# Peer review of "Tuning the Doping Ratio and Phase Transition Temperature of VO2 Thin Film by Dual-Target Co-Sputtering"

_nanomaterials, 2019, doi:10.3390/nano9060834_

Reviewer 1 Report

This paper contains no error analysis whatsoever.  There are statements about numerical results that have no error bars.  No statement about the number of trials is given.  Compositions, results etc are given with high precision and significant digits without justification.

This paper MUST include error analysis to be published.  The presentation of a method to make a specific material must contain the limits on the fabrication method.

Some Details:

line 29 and others.  Do not use adjectives such as "huge" "large" etc.  They have no place in scientific writing.  If you have a range, give the numbers and a reference. Same for lines 38 "very low", 41 "very difficult"..and so on.

Line 44: titles are not used in references.  Delete "Dr,"

line 54: you do not include any cost estimates or cost data..  Delete "cost is too high".

line 61: "room temperature is not a scientific concept.  Use the actual temperature.

line 78 "planar" ?

line 87: "coating power"?  "500 W with increasing step of 100 W per minute"  What does this mean? Was the power increased?

There is a paper that suggest the sputtering power changes the transition temperature.

What evidence do you have the sputtering power does not cause the change in transition temperature?

Infrared-light switching in highly oriented VO2 films on ZnO-buffered glasses with controlled phase transition temperatures

By:Hoshino, H (Hoshino, Hiroaki)[ 1 ] ; Okimura, K (Okimura, Kunio)[ 1 ] ; Yamaguchi, I (Yamaguchi, Iwao)[ 2 ] ; Tsuchiya, T (Tsuchiya, Tetsuo)[ 2 ]

SOLAR ENERGY MATERIALS AND SOLAR CELLS

Volume: 191

Pages: 9-14

DOI: 10.1016/j.solmat.2018.10.022

lines 93 to 98  these are the expected compositions, were they verified?

line 114 ???  "The total....speed". Makes no sense.

line 129.  It is a good practice to separate results from discussion.  The discussion on lines 130 onward is speculation and does not belong in a results section. 

Figure 2: The scale bars are too small to be seen in the figures.

Figure 3:  reference marks for peaks of standard compounds are usually shown to orient the

reader.  The V6O11 peak is small, are there any other peaks?  How are the compositions determined to such precision? V0.98 ?? W.02?? +/- % ? Also in line 149.

Line 151; how are the percentages calculated, what error?

Figure 4: Error bars? Significant figures? How reproducible is this data.

line 174 : How did you determine the W0.01?

Figure 6 Error bars? in b).

line 219. You have made no cost estimates. Delete this claim.

Author Response

Dear Reviewer,

Thank you very much for carefully review and constructive comments on our manuscript entitled “Tuning the doping ratio and phase transition temperature of VO2 thin film by dual-target co-sputtering” (Manuscript ISSN 2079-4991). We studied the comments carefully and tried our best to revise and improve the manuscript according to your comments. The point to point response to your comments are listed as following:

Point 1: There are statements about numerical results that have no error bars. No statement about the number of trials is given.  Compositions, results etc are given with high precision and significant digits without justification. This paper MUST include error analysis to be published.  The presentation of a method to make a specific material must contain the limits on the fabrication method. 

Response 1: Sorry for negligence of error analysis. We have modified them in the text according to your comments. For example, we analyzed the signal-to-noise ratio (SNR) of the test spectrum for the test error. It is larger than 10000 for all the spectral measurements and is good enough. The diffraction peaks of XRD results agree with the standard cards well. The largest deviation of 1.4% comes from the first peak of V2O5 at 12.404 degree from the standard card. The error of sputtering power deviation is less than 340 V ±5 V and corresponding to maximum error of 1.5%. All mentioned above have also been added into the corresponding parts of the manuscript.

Point 2: Line 29 and others.  Do not use adjectives such as "huge" "large" etc. They have no place in scientific writing.  If you have a range, give the numbers and a reference. Same for lines 38 "very low", 41 "very difficult"..and so on. 

Response 2: We have made the corrections according to your comments.

Point 3: Line 44: titles are not used in references.  Delete "Dr," 

Response 3: We deleted "Dr," according to your comment.

Point 4: Line 54: you do not include any cost estimates or cost data..  Delete "cost is too high". 

Response 4: We deleted "cost is too high" according to your comment.

Point 5: Line 61: "room temperature is not a scientific concept.  Use the actual temperature. 

Response 5: The room temperature is usually default as 25℃. But in the manuscript, we use it to express the meaning of “without heating” during the sputtering process. So we substitute “room temperature” for “without heating”.

Point 6: Line 78 "planar" ? 

Response 6: We substitute “plane” for “planar” throughout the manuscript according to your comment.

Point 7: Line 87: "coating power"?  "500 W with increasing step of 100 W per minute"  What does this mean? Was the power increased? 

There is a paper that suggest the sputtering power changes the transition temperature.

What evidence do you have the sputtering power does not cause the change in transition temperature? (Infrared-light switching in highly oriented VO2 films on ZnO-buffered glasses with controlled phase transition temperatures. By:Hoshino, H (Hoshino, Hiroaki)[ 1 ] ; Okimura, K (Okimura, Kunio)[ 1 ] ; Yamaguchi, I (Yamaguchi, Iwao)[ 2 ] ; Tsuchiya, T (Tsuchiya, Tetsuo)[ 2 ].SOLAR ENERGY MATERIALS AND SOLAR CELLS.Volume: 191,Pages: 9-14,DOI: 10.1016/j.solmat.2018.10.022) 

Response 7: Sorry for the ambiguous expression. What we want to express is that the sputtering power is set as 500 W when coating the V film, which is increased from 0 W to 500 W at a rate of 100 W per minute before coating. The sputtering power does not change during the coating process. Since the fabrication approach is different from that of the reviewer provided, which contains two procedures for fabrication of VO2 film. V metal film is sputtered firstly and then annealed and oxidized to be VO2 film. What sputtered is V metal film (not VO2 film), whose sputtering power does not influence the transition temperature at all. Therefore, our fabrication approach is easy to be controlled and the repeatability is higher than those approaches sputtering the VO2 film directly.

Point 8: Lines 93 to 98  these are the expected compositions, were they verified?

Response 8: The compositions are expected and were not verified. We inserted "supposed to be" between "the deposition ratio of V and W-doped V is" and "2:0, 2:1, 2:2, 1:2, 0:2".

Point 9: Line 114 ???  "The total....speed". Makes no sense.

Response 9: It should be misunderstood here due to the vague expression of “times” and “time”. For the sputtering system we used, the substrate was put on the conveyor and moved backward and forward underneath the targets to increase the homogeneity during sputtering. We defined that when the substrate moves backward and forward underneath the targets once as one sputtering cycle. Therefore, the thickness of each sputtering cycle was deduced from the SEM measured thickness of five sputtering cycles. We modified the related expression in the context. 

Point 10: Line 129.  It is a good practice to separate results from discussion.  The discussion on lines 130 onward is speculation and does not belong in a results section. 

Response 10: Thank you for your suggestion for dealing with results and discussion. However, it is very difficult to separate results from discussion, since the results need proper explanation or reasonable speculation after them for readers to better understand them.

Point 11: Figure 2: The scale bars are too small to be seen in the figures. 

Response 11: We have re-drawn Figure 2 with larger scale bars according to your suggestion.

Point 12: Figure 3:  reference marks for peaks of standard compounds are usually shown to orient thereader.  The V6O11 peak is small, are there any other peaks?  How are the compositions determined to such precision? V0.98 ?? W.02?? +/- % ? Also in line 149.

Response 12: We have re-drawn Figure 3 according to your comment. No peaks other than V6O11 and V2O5 were found from the XRD analysis. The composition of V0.98W0.02O2 was ensured from the target with V: W ratio of 98:2 we ordered. 

Point 13: Line 151; how are the percentages calculated, what error?

Response 13: The percentages of W in VO2 were calculated from the sputtering power ratio of V target and W doped V target which were set as 500 W :0 W, 500 W :250 W, 500 W :500 W, 250 W :500 W, 0 W :500 W, respectively. The error comes from the power deviation which is less than 340 V±5 V and corresponding to maximum of 1.5%.

Point 14: Figure 4: Error bars? Significant figures? How reproducible is this data.

Response 14: Signal to noise ratio (SNR) is used to evaluate the error level of spectrum rather than error bars, which is a common way in the field. The larger of SNR, the smaller of error. It is larger than 10000 for all the spectral measurements and good enough.

Point 15: line 174 : How did you determine the W0.01?

Response 15: The W0.01 was deduced from the sputtering power ratio of V target and W doped V target, as mentioned in point 13.

Point 16: Figure 6 Error bars? in b).

Response 16: Since the phase transition temperature are calculated from the spectra and the main purpose of Figure 6 (b) is to provide the change and reveal the trend of phase transition temperature with W doping ratio, the error bars are difficult and not necessary to be provided in the figure.

Point 17: Line 219. You have made no cost estimates. Delete this claim. 

Response 17: Although we have not made cost estimates, we fabricated a series of VO2 samples with different doping ratio without adding any targets. At least the cost of three targets with doping ratio of 0.5%, 1% and 1.5% and size of 300 mm ×140 mm has been saved. More cost can be saved if more samples with other different doping ratio to be fabricated. Therefore, the claim of cutting cost using co-sputtering approach to fabricate different doping ratio is reasonable.

We tried our best to modify and improve the manuscript according to your comments. And we used the “Track Changes” function in Microsoft Word to mark the revision, so that all the changes can be easily noticed by the editors and reviewers. 

We appreciate for your constructive and helpful comments on our manuscript, and hope that the correction and response will meet your requirements.

Reviewer 2 Report

The authors have described a way to fabricate thin films of VO2 by doping as a result of a cosputtering process. The experiment and the deposition/doping process are well described. I would like to suggest to the authors to think about the rate of deposition from each target which sometimes can be different than the rate of cumulative deposition (this might need something profilometry and EDS to compute the exact rate during cosputtering).

Author Response

Dear Reviewer,

Thank you very much for carefully review and constructive comments on our manuscript entitled “Tuning the doping ratio and phase transition temperature of VO2 thin film by dual-target co-sputtering” (Manuscript ISSN 2079-4991). We studied the comment carefully and tried our best to revise and improve the manuscript according to your comment. The response to your comment is listed as following:

Point 1: The authors have described a way to fabricate thin films of VO2 by doping as a result of a cosputtering process. The experiment and the deposition/doping process are well described. I would like to suggest to the authors to think about the rate of deposition from each target which sometimes can be different than the rate of cumulative deposition (this might need something profilometry and EDS to compute the exact rate during cosputtering). 

Response 1: It is a very good question. In fact, the rate of deposition from each target sometimes might be different from the rate of cumulative deposition. However, the instantaneous deposition rate of each target is very difficult to be accurately determined even by profilometry and EDS. Therefore, the average rate for cumulative deposition is a common way to deal with such case and agrees with experimental results well.

Many thanks for your comment.

We tried our best to modify and improve the manuscript according to your comment. And we used the “Track Changes” function in Microsoft Word to mark the revision, so that all the changes can be easily noticed by the editors and reviewers.

We appreciate for your constructive and helpful comments on our manuscript, and hope that the correction and response will meet your requirement.

Reviewer 3 Report

The authors did a good job for this manuscript. However, I do have some concerns and questions about the manuscript.

1- The abstract doesn't have enough information to picture the whole manuscripts such as including some results and conclusion. 

2- Line 41 # "[11]. However, the films are deposited at a temperature higher than 450℃." What is the exact deposition temperature? Higher than 450 Celcius could be 454 Celcius or could be 600 Celcius. It needs to be more clear.

3- Line 59 # the sputtering system and targets weren't detailed such as brand name, model name... Line 83# "the chamber of magnetron sputtering coating system..." should given the brand name. 

4- Line 84 # the authors mentioned the substrate never been in an amorphous glass and then given one article with the possible both type of the glass in the introduction part. The question is what the substrate type is here? If it is glass, what type of glass is? Amorphous or others? 

5- Line 84  # the authors wrote " to lower than 9.9×10-4 Pa ".. But, what is the lowest vacuum pressure. Why didn't mention the exact pressure? Is it really difficult to measure and record it? If then how can you say that these results can be reproducible? 

6- Line 87 #, in this line, the working pressure [I believed]  mentioned that 2.1×10-2 Pa. If this the working pressure, and if they are using a co-sputtering system, how could solve the flux issue such very high pressure? Their method and their atomic ratio calculations can be understood/ acceptable during low pressure.

7- Line 89#  "The targets were sputtered for 30 minutes before sputtering onto the substrate. The deposition temperature was room temperature and did not need to heat the substrate."

I do have a couple of questions here; For the first sentence, How did you handle the pre-deposition inside the chamber?  Is there any sputter gun shutter to protect the sample? If there is not any shutter, how do you protect the substrate from the pre-deposition flux? 

Second questions, Have you able to measure the temperature of the target during or right after the deposition via laser temp gun? How long did you deposit? After 15 minutes later, the target's temperature can be increased 15-25 Celcius which can affect the sample. Have you considered this? How did you solve this issue?

8- Line 99 #  The authors mentioned the resultants (I think they are the deposited substrates). Are these resultants were directly put inside the annealing furnace or did they wait outside for a while? If they waited outside for a long time, how did you handle the oxidize issue?

9- Line 114 # 2n-1 thicknesses can be realized by only n times of sputtering," Where/how did you find this formula to calculate the thickness? 

10- Line 137# Figure 2; It should be better to name each of the images such as a, b, c and d. It is not clear which one is annealed images which one before annealing. 

11- Line 142# Figure 3; It should be better to define the type of glass used inside the deposition. 

12- Line 150 # what is the phase transition temperature of this deposition?

Last but not least, for the conclusion section, the atomic weight samples weren't mentioned!

Again, thanks for the good work and I believe it will be more strong paper after answering all of these questions. 

Good Luck!

Author Response

Dear Reviewer,

Thank you very much for carefully review and constructive comments on our manuscript entitled “Tuning the doping ratio and phase transition temperature of VO2 thin film by dual-target co-sputtering” (Manuscript ISSN 2079-4991). We studied the comments carefully and tried our best to revise and improve the manuscript according to your comments. The point to point response to your comments are listed as following:

 Point 1: The abstract doesn't have enough information to picture the whole manuscripts such as including some results and conclusion.

 Response 1: Many thanks for your helpful comment on the abstract and conclusion. We added "A series of samples with W doping ratio of 0%, 0.5%, 1%, 1.5% and 2% have been fabricated just by sputtering V films with power of pure and 2% W-doped V targets from 500 W :0 W, 500 W :250 W, 500 W :500 W, 250 W :500 W to 0 W :500 W respectively and then annealed in oxygen atmosphere to form VO2. The XRD results of both pure and W doped VO2 samples reveal that VO2 forms and is the main component after annealing." and ", with W doping ratio from 0% to 2%" into the abstract. We also added some sentences of "It is 64.5 for the film of pure VO2 film without doping and reduces with the increase of doping ratio. When the W doping ratio is 0.5%, the phase transition temperature decreases 9.5 to 55. It is lowered to 45 when the doping ratio increases to 1%. The phase transition temperature is further reduced to 42.5 when the W content increases to 1.5%. The phase transition temperature is down to 36 for 2% doping ratio." into the conclusion.

 Point 2: Line 41 # "[11]. However, the films are deposited at a temperature higher than 450℃." What is the exact deposition temperature? Higher than 450 Celcius could be 454 Celcius or could be 600 Celcius. It needs to be more clear.

 Response 2: Sorry for the improper expression. We changed it to 400 which is mentioned in reference [5], according to your comment.

 Point 3: Line 59 # the sputtering system and targets weren't detailed such as brand name, model name... Line 83# "the chamber of magnetron sputtering coating system..." should given the brand name.

 Response 3: The sputtering system used is a home-made large area magnetron sputtering system, whose photo (as shown in the following picture) and detailed information have already been reported in other place [12]. The targets we used were planar metal targets with size of 300 mm ×140 mm which were mentioned in 81# of the manuscript. If the picture below is not displayed, please check the WORD version.

Point 4: Line 84 # the authors mentioned the substrate never been in an amorphous glass and then given one article with the possible both type of the glass in the introduction part. The question is what the substrate type is here? If it is glass, what type of glass is? Amorphous or others?

Response 4: This method is valid for fabrication of VO2 thin films on both amorphous and crystalline substrates, such as glass and sapphire. The substrate we used here is amorphous glass of K9. We added it into the corresponding part of the manuscript.

Point 5: Line 84  # the authors wrote " to lower than 9.9×10-4 Pa ".. But, what is the lowest vacuum pressure. Why didn't mention the exact pressure? Is it really difficult to measure and record it? If then how can you say that these results can be reproducible?  

Response 5: Deposition in vaccum is to avoid the collision of sputtered atoms with molecules in air and influence the quality of sputtered film. 9.9×10-4 Pa is already good enough for depositing high quality thin film with good reproducibility. The vaccum pressure will decrease further with the pumping time. The lowest vacuum pressure can reach 5×10-4 Pa which need almost a day to reach the equilibrium. Since the reproducibility is already good for sputtering with vaccum lower than 9.9×10-4 Pa, it does not need to pumping down to the lowest vaccum pressure by taking the efficiency into account.

Point 6: Line 87 #, in this line, the working pressure [I believed]  mentioned that 2.1×10-2 Pa. If this the working pressure, and if they are using a co-sputtering system, how could solve the flux issue such very high pressure? Their method and their atomic ratio calculations can be understood/ acceptable during low pressure.

Response 6: Yes, you are right. 2.1×10-2 Pa is the working pressure when Argon gas with flow rate of 50 sccm was injected into the sputtering chamber as working gas. The flux issue can be controlled by a precise flowmeter. Argon is necessary in the experiment to sputter out the V and W:V atoms from the targets. It is a routine process and the working pressure can only be kept at this level when working gas injected into the sputtering chamber. Thank you.

Point 7: Line 89#  "The targets were sputtered for 30 minutes before sputtering onto the substrate. The deposition temperature was room temperature and did not need to heat the substrate."

I do have a couple of questions here; For the first sentence, How did you handle the pre-deposition inside the chamber?  Is there any sputter gun shutter to protect the sample? If there is not any shutter, how do you protect the substrate from the pre-deposition flux?

Second questions, Have you able to measure the temperature of the target during or right after the deposition via laser temp gun? How long did you deposit? After 15 minutes later, the target's temperature can be increased 15-25 Celcius which can affect the sample. Have you considered this? How did you solve this issue?

Response 7: Thank you for your comments. Maybe it's better for you to understand them with the system photo we provided in Point 3. For the first question, there is an independent chamber on the left side (see the photo) separated by a vaccum valve. When the sputtering chamber is pre-sputtered for 30 minutes, the sample is separated in the independent chamber. When the pre-sputtering is finished, the valve is opened. And the sample moves backward and forward within the sputtering chambers on the right side to ensure the uniformity of the film.

For the second question, the deposition principle has been shown in the schematic diagram of Figure 1. You are right. Although without heating during the deposition process, the temperature will increase gradually due to the sputtering energy. But the influence is very small and can be neglected for the sputtering time is short. Furthermore, what sputtered is V metal film (not VO2 film) which is stable and the increased temperature during sputtering has little effect on the sample. Therefore, our fabrication approach is very easy to be controlled and the repeatability is higher than those approaches sputtering the VO2 film directly. It is a promising approach for large-scale production of sputtering.

Point 8: Line 99 #  The authors mentioned the resultants (I think they are the deposited substrates). Are these resultants were directly put inside the annealing furnace or did they wait outside for a while? If they waited outside for a long time, how did you handle the oxidize issue?

Response 8: You are right that we mentioned the resultants are the deposited substrates. After finishing the coating, the samples were directly placed into a large-diameter vacuum annealing furnace for annealing process which is near the sputtering system and in the same room.

Point 9: Line 114 # 2n-1 thicknesses can be realized by only n times of sputtering," Where/how did you find this formula to calculate the thickness?  

Response 9: Sorry for writing the formula by mistake. The correct formula is 2n-1. By n times of combinatorial depositions, 2n deposition thicknesses can be formed on the substrate, including 0 nm deposition one. Therefore, 2n-1 thicknesses can be obtained after n times of combinatorial depositions. Please see the detailed information about the combinatorial deposition technique reported in reference [22].

Point 10: Line 137# Figure 2; It should be better to name each of the images such as a, b, c and d. It is not clear which one is annealed images which one before annealing.

Response 10: Thank you for your suggestion to improve our manuscript. We have modified Figure 2 according to your comment.

Point 11: Line 142# Figure 3; It should be better to define the type of glass used inside the deposition.

Response 11: According to your suggestion, we added the type information of glass used during the deposition is K9 glass, and mentioned it in the manuscript.

Point 12: Line 150 # what is the phase transition temperature of this deposition?

Response 12:  The phase transition temperature of VO2 is the temperature when its phase transition occurs between the insulation phase and metal phase. It can be obtained from the optical thermal hysteresis loop of VO2 and described in the manuscript.

Point 13: Last but not least, for the conclusion section, the atomic weight samples weren't mentioned!

Response 13: In the conclusion section, we added the change of phase transition temperature of VO2 with different W doping ratio. We added some sentences of "It is 64.5 for the film of pure VO2 film without doping and reduces with the increase of doping ratio. When the W doping ratio is 0.5%, the phase transition temperature decreases 9.5 to 55. It is lowered to 45 when the doping ratio increases to 1%. The phase transition temperature is further reduced to 42.5 when the W content increases to 1.5%. The phase transition temperature is down to 36 for 2% doping ratio." into the conclusion.

We tried our best to modify and improve the manuscript according to your comments. And we used the “Track Changes” function in Microsoft Word to mark the revision, so that all the changes can be easily noticed by the editors and reviewers.

We appreciate for your constructive and helpful comments on our manuscript, and hope that the correction and response will meet your requirements.

Reviewer 4 Report

1. Introduction

 1.1. The authors' statement «It is also valid for other doping elements and is a promising approach for large-scale production of sputtering to cut the cost» is incorrect in its second part, since a magnetron with a vanadium target is not required in a "large-scale production". The authors indeed offer a new method for studying the technology of (WO3)1-x(VO2)x film. The optimum concentration of tungsten should be determined in two-magnetron study. At this concentration, the transition temperature is minimal. In production, only one magnetron with a composite target having a given concentration of tungsten should be used to reduce costs.

1.2. The authors don’t indicate more recent publications than [17] devoted to a similar problem, in which a decrease in the phase transition temperature is obtained. For example, [1] where dopant reduce the phase transition temperature of VO2 thin films to 50°C or [2], where a dopant reduces the phase transition temperature of VO2 thin films to 21.9°C, or [3], where a dopant reduces the phase transition temperature of VO2 thin films to 28°C.

 1. Kang S.H. Han S.H., Park S.J., Kim H., Yang W.S. Optical properties of VO2 thin film deposited on F: SnO2 substrate for smart window application Korean Journal of Materials Research 23 (2013) 215–218.

2. Chen, S., Zhang, L. F., Xiao, S. J., & Dong, C. Z. Structure Characterization of W-Doped Vanadium Oxide Thin Films Prepared by Reactive Magnetron Sputtering. Adv. Mater. Res. 690-693 (2013) 1694–1697.

3. Li J.-H., Yuan N.-Y., Xie T.-B., Dan D.-D. Preparation of polycrystalline V0.97W0.03O2 thin films with ultra high TCR at room temperature Wuli Xuebao/Acta Phys. Sinica 56 (2007) 1790–1795.

 2. Experimental procedure

 2.1. The title of Figure 1. «Schematic diagram of fabricating W-doped VO2 thin films …» should be written correctly: «Schematic diagram of fabricating W-doped V thin films …» 

 2.2. There is no information about the study on the influence of oxygen pressure on the properties of films during annealing.

 3. Results and discussion

 3.1. The authors don’t analyze the physical causes of the transition temperature decrease.

 3.2. The authors don’t give a comparative assessment of the results with the results of other authors.

 3.3. The authors don’t comment on the reasons for obtaining a higher transition temperature than that achieved by other authors.

Author Response

Dear Reviewer,

Thank you very much for carefully review and constructive comments on our manuscript entitled “Tuning the doping ratio and phase transition temperature of VO2 thin film by dual-target co-sputtering” (Manuscript ISSN 2079-4991). We studied the comments carefully and tried our best to revise and improve the manuscript according to your comments. The point to point response to your comments are listed as following:

Point 1: The authors' statement «It is also valid for other doping elements and is a promising approach for large-scale production of sputtering to cut the cost» is incorrect in its second part, since a magnetron with a vanadium target is not required in a "large-scale production". The authors indeed offer a new method for studying the technology of (WO3)1-x(VO2)x film. The optimum concentration of tungsten should be determined in two-magnetron study. At this concentration, the transition temperature is minimal. In production, only one magnetron with a composite target having a given concentration of tungsten should be used to reduce costs. 

Response 1: You are right. It needs only a composite target with given concentration of tungsten when in production to reduce costs. However, new targets are necessary if VO2 products with other phase transition temperature or doping ratios need to be fabricated to match different demands, even on the research stage. In these cases, many targets with different doping ratio are needed for conventional method. The main purpose of this work is to introduce a method for tuning doping ratio and phase transition temperature of VO2 thin film by dual-target co-sputtering. The optimum concentration of tungsten is not the work we want to do in this manuscript, which need clearly definition for the lowest phase transition temperature or largest modulation ability on infrared transmission and so on.

Point 2: The authors don’t indicate more recent publications than [17] devoted to a similar problem, in which a decrease in the phase transition temperature is obtained. For example, [1] where dopant reduce the phase transition temperature of VO2 thin films to 50°C or [2], where a dopant reduces the phase transition temperature of VO2 thin films to 21.9°C, or [3], where a dopant reduces the phase transition temperature of VO2 thin films to 28°C.

( 1. Kang S.H. Han S.H., Park S.J., Kim H., Yang W.S. Optical properties of VO2 thin film deposited on F: SnO2 substrate for smart window application Korean Journal of Materials Research 23 (2013) 215–218.

2. Chen, S., Zhang, L. F., Xiao, S. J., & Dong, C. Z. Structure Characterization of W-Doped Vanadium Oxide Thin Films Prepared by Reactive Magnetron Sputtering. Adv. Mater. Res. 690-693 (2013) 1694–1697.

3. Li J.-H., Yuan N.-Y., Xie T.-B., Dan D.-D. Preparation of polycrystalline V0.97W0.03O2 thin films with ultra high TCR at room temperature Wuli Xuebao/Acta Phys. Sinica 56 (2007) 1790–1795.)

Response 2: Thank you for providing us the recent related publications. We added them to related part of the manuscript and cited as references.

Point 3: The title of Figure 1. «Schematic diagram of fabricating W-doped VO2 thin films …» should be written correctly: «Schematic diagram of fabricating W-doped V thin films …»  

Response 3: Thank you for your comment. We corrected it in the manuscript.

Point 4: There is no information about the study on the influence of oxygen pressure on the properties of films during annealing.

Response 4: Thank you for your comment. The influence of oxygen pressure on the properties of films during annealing is really very important for our method to fabricate VO2 thin films. We have already studied it systematically and published in other place as reference [24].  Therefore, we did not present those results here.

Point 5: The authors don’t analyze the physical causes of the transition temperature decrease.

Response 5: The reason why doping affects the phase transition temperature of VO2 crystal has not been uniformly explained by theoretical theory. According to the existing theories, the phase transition temperature of VO2 film reduced by W doping may be due to the substitution of W6+ to V4+ in d orbital. The introduction of excess electrons reduces the band gap of the d orbit and reduces the phase transition temperature, which can be seen in reference [19].  We added the explanation to the related part of the manuscript.

Point 6: The authors don’t give a comparative assessment of the results with the results of other authors.

Response 6: We have mentioned the results of other researchers for comparison in the manuscript, which can be seen between the  line 211#-215# in the text as “ For the transition temperature, it is reduced to 55 and 35 with VO2-W-1% and VO2-W-2% respectively which was reported by Liu et al. using the aqueous sol-gel method [13]. Zhang et al. reported a decrease to 55.3 and 45.7 respectively with VO2-W-1% and VO2-W-2% synthesized by a facile hydrothermal approach and subsequent calcination [14].

Point 7: The authors don’t comment on the reasons for obtaining a higher transition temperature than that achieved by other authors.

Response 7: The reason we get higher transition temperature than that achieved by other authors may be due to differences in the preparation methods. For example, Liu et al. using the aqueous sol-gel method. Zhang et al. using a facile hydrothermal approach and subsequent calcination.

We tried our best to modify and improve the manuscript according to your comments. And we used the “Track Changes” function in Microsoft Word to mark the revision, so that all the changes can be easily noticed by the editors and reviewers.

We appreciate for your constructive and helpful comments on our manuscript, and hope that the correction and response will meet your requirements.

Round  2

Reviewer 1 Report

The manuscript is improved.  The questions of statistical analysis have not been answered by the authors.

How many times was each composition evaluated for the shift in transition temperature?

The ultimate question is:

IF I sputter composition x n times, how many times will I get transition temperature y +/-z?

This is the equivalent of Figure 6 b, with the number of trials and a standard deviation for each.

This must be included.

Figure 3, the XRD, is unacceptable as shown.  The identification of the peaks is

random.  The V6O11 identification is unlikely. There are 3 equally intense peaks between 25 and 31 degrees.  Only one is observed, and it is an unusual orientation.  The peak at 28 degrees is already in the VO2 pattern.  These inconsistencies must be explained. 

Author Response

Dear Reviewer,

Thank you very much for carefully review and constructive comments on our manuscript entitled “Tuning the doping ratio and phase transition temperature of VO2 thin film by dual-target co-sputtering” (Manuscript ID: nanomaterials-490208). We studied the comments carefully and tried our best to revise and improve the manuscript according to your comments. The point to point response to your comments are listed as following:

Point 1: How many times was each composition evaluated for the shift in transition temperature? 

The ultimate question is:

IF I sputter composition x n times, how many times will I get transition temperature y +/-z?

This is the equivalent of Figure 6 b, with the number of trials and a standard deviation for each.

This must be included.

Response 1: Sorry for not catching the exact meaning in your previous comments. The experiments of different doping ratio have been repeated three times. The deviation of phase transition temperature has been marked in Figure 6(b), according to your request. It is no more than 3.03% and the largest one comes from the samples of 1.5% W-doped VO2 films. Additionally, we changed all the phase transition temperatures mentioned in the manuscript to the average results of repeated experiments as follows. Many thanks for your patience and giving us chance to improve it.  If the picture below is not displayed, please open the word document to view.

Point 2: Figure 3, the XRD, is unacceptable as shown.  The identification of the peaks is

random.  The V6O11 identification is unlikely. There are 3 equally intense peaks between 25 and 31 degrees.  Only one is observed, and it is an unusual orientation.  The peak at 28 degrees is already in the VO2 pattern.  These inconsistencies must be explained.

Response 2: Thank you very much for pointing out the mistake of peak identification. It is really unlikely to identify the peak of 29.14 degree as V6O11 (-1-17).  We re-identify the peaks and redraw Figure 3 again. The two main peaks at 28 degree and 29 degree are identified as VO2 and V2O5, respectively. The small deviation of V2O5 at 29 degree may be ascribed to the difference from different fabrication methods. Different orientations observed should be ascribed to the fabrication of thin films on amorphous substrates of K9 glass. This comment is very important for us. Many thanks again for help us avoiding a mistake in our manuscript.

We tried our best to modify and improve the manuscript according to your comments. And we used the “Track Changes” function in Microsoft Word to mark the revision, so that all the changes can be easily noticed by the editors and reviewers.

We appreciate for your constructive and helpful comments on our manuscript, and hope that the correction and response will meet your requirements.

Reviewer 3 Report

Dear Authors,

Thanks for the updates. I do have one question and one concern about the manuscript.

Concern; I couldn't see the update on Figure 2 such as a,b,c and etc didn't define yet. 

Question; line #104; "of V and W-doped V is supposed to be 2:0, 2:1, 2:2, 1:2, 0:2, respectively. Therefore, the amount of W" What does it mean "supposed to be"? Didn't you calculate and measure the ratios?

Very appreciated!

Thanks 

Author Response

Dear Reviewer,

Thank you very much for carefully review and constructive comments on our manuscript entitled “Tuning the doping ratio and phase transition temperature of VO2 thin film by dual-target co-sputtering” (Manuscript ID: nanomaterials-490208). We studied the comments carefully and tried our best to revise and improve the manuscript according to your comments. The point to point response to your comments are listed as following:

Point 1: I couldn't see the update on Figure 2 such as a,b,c and etc didn't define yet. 

Response 1: We corrected Figure 2 and the corresponding figure caption according to your comment. It is “(a) Pure V film before annealing. (b) 2% W-doped V film before annealing. (c) Pure V film after annealing. (d) 2% W-doped V film after annealing.”, which can also be seen in the manuscript.

Point 2: line #104; "of V and W-doped V is supposed to be 2:0, 2:1, 2:2, 1:2, 0:2, respectively. Therefore, the amount of W" What does it mean "supposed to be"? Didn't you calculate and measure the ratios?

Response 2: Sorry, we used an improper word to describe it due to English is non-native language and not good at it. In fact, what we want to express is the ratios are calculated from the sputtering power ratios. Since the sputtering speed is proportional to the sputtering power and controls the sputtering rate, the sputtered amount ratio from each target was 2:1, 2:2 and 1:2 when the power ratio of the two targets is 500 W:250 W, 500 W:500 W and 250 W:500 W. We changed “supposed” to “calculated” and revised related part as follows: “the deposition ratio of V and W-doped V is calculated to be 2:1, 2:2 and 1:2 for power ratios of 500 W :250 W, 500 W :500 W and 250 W :500 W, respectively. For the samples with powers of 500 W :0 W and 0 W :500 W correspond to the pure and 2% W-doped ones, respectively.”, which can also be seen in the manuscript. If the picture below is not displayed, please open the word document to view.

We tried our best to modify and improve the manuscript according to your comments. And we used the “Track Changes” function in Microsoft Word to mark the revision, so that all the changes can be easily noticed by the editors and reviewers.

We appreciate for your constructive and helpful comments on our manuscript, and hope that the correction and response will meet your requirements now.

Reviewer 4 Report

Dear Colleagues,

 The manuscript has been improved significantly. Nevertheless, the Responses 6 and 7 should be refined.

Response 6: We have mentioned the results of other researchers for comparison in the manuscript, which can be seen between the  line 211#-215# in the text as “ For the transition temperature, it is reduced to 55℃ and 35℃ with VO2-W-1% and VO2-W-2% respectively which was reported by Liu et al. using the aqueous sol-gel method [13]. Zhang et al. reported a decrease to 55.3℃ and 45.7℃ respectively with VO2-W-1% and VO2-W-2% synthesized by a facile hydrothermal approach and subsequent calcination [14]”.

Comment. The Authors don’t provide a comparison with better results obtained in [18] and [19].

 Response 7: The reason we get higher transition temperature than that achieved by other authors may be due to differences in the preparation methods. For example, Liu et al. using the aqueous sol-gel method. Zhang et al. using a facile hydrothermal approach and subsequent calcination.

Comment. It is possible that the new Author's method based on co-sputtering and annealing in oxygen atmosphere of a W-V metal film, cannot reduce the transition temperature to 20 ℃, in principle.

It can be caused by the poor controllability of the thermal oxidation reaction. The phase boundary of W-V-O and W-V moves from the film surface towards the substrate due to the diffusion of components and chemical reaction. In this case, the preferential growth of the VO2 component, which has a clear dielectric-metal thermal phase transition, is not ensured. XRD curves in Figure 3 show that the films contain V2O5, which has no such properties.

As a result, doping a VO2 film with tungsten reduces the transition temperature, and the appearance of a V2O5 component in a W-V-O solid solution increases it. In the manuscript, there is no discussion of such opportunities.

Author Response

Dear Reviewer,

Thank you very much for carefully review and constructive comments on our manuscript entitled “Tuning the doping ratio and phase transition temperature of VO2 thin film by dual-target co-sputtering” (Manuscript ID: nanomaterials-490208). We studied the comments carefully and tried our best to revise and improve the manuscript according to your comments. The point to point responses to your comments are listed as following:

Point 1:

Response 6: We have mentioned the results of other researchers for comparison in the manuscript, which can be seen between the  line 211#-215# in the text as “For the transition temperature, it is reduced to 55℃ and 35℃ with VO2-W-1% and VO2-W-2% respectively which was reported by Liu et al. using the aqueous sol-gel method [13]. Zhang et al. reported a decrease to 55.3℃ and 45.7℃ respectively with VO2-W-1% and VO2-W-2% synthesized by a facile hydrothermal approach and subsequent calcination [14]”.

Comment. The Authors don’t provide a comparison with better results obtained in [18] and [19].

Response 1: The reason why we chose the phase transition temperatures of W-doped VO2 films in reference [13] and [14] for comparison is they provided exactly the same W doping ratio (1% and 2%) results as ours. Since different doping ratio results in different phase transition temperature (inversely proportional to doping ratio of W), the results the reviewer mentioned of references [18] and [19] can’t be used for comparison directly because the better result obtained in [19] is with 3% W-doped VO2 doping ratio. The phase transition temperature should decrease further if we fabricate W doped VO2 thin film with higher doping ratio. We added a sentence to mention the result in [19] as you request as follows: “Li et al. used ion beam enhanced deposition and post-annealing techniques to prepare 3% W-doped VO2 on SiO2 substrates which reduced the phase transition temperature of VO2 thin films to 28°C [19].”

  As for the result in reference [18] you mentioned, the phase transition temperature is not measured by a common method and does not provide the doping ratio, which result in the difficulty for comparison with our results. Therefore, we didn’t mention it in the context. 

Point 2:

Response 7: The reason we get higher transition temperature than that achieved by other authors may be due to differences in the preparation methods. For example, Liu et al. using the aqueous sol-gel method. Zhang et al. using a facile hydrothermal approach and subsequent calcination.

Comment. It is possible that the new Author's method based on co-sputtering and annealing in oxygen atmosphere of a W-V metal film, cannot reduce the transition temperature to 20℃, in principle.

It can be caused by the poor controllability of the thermal oxidation reaction. The phase boundary of W-V-O and W-V moves from the film surface towards the substrate due to the diffusion of components and chemical reaction. In this case, the preferential growth of the VO2 component, which has a clear dielectric-metal thermal phase transition, is not ensured. XRD curves in Figure 3 show that the films contain V2O5, which has no such properties.

As a result, doping a VO2 film with tungsten reduces the transition temperature, and the appearance of a V2O5 component in a W-V-O solid solution increases it. In the manuscript, there is no discussion of such opportunities.

Response 2: Sorry, we did not read the references you provided last time carefully and did not notice these results are based on different doping ratio. In fact, for the 2% W-doping case, the phase transition temperature of samples fabricated by our method can be reduced down to 36.5 similar to other methods such as 35 reported in reference [13], or even lower than that of 45.7 reported in reference [14]. It should be able to be reduced down to 20 if we use target with larger doping ratio, according to the decrease trend with doping ratio revealed by Soltani et. al. [18] which published in Applied Physics Letters.

As you pointed out, the controllability of the thermal oxidation reaction is really poor for conventional methods, because the annealing environment, atmosphere and parameters for thermal oxidation reaction is complex and they will influence the formation of vanadium oxides. Therefore, we developed a new approach to fabricate the VO2 thin films by sputtering V metal film and then annealed in pure and precisely controlled oxygen atmosphere by evacuating the chamber of furnace firstly and then annealed with pure oxygen. The sputtering process of V film and annealing process with pure and precisely controlled oxygen atmosphere are more stable than conventional fabrication methods. Therefore, the controllability and repeatability of our method is good.

As for the influence mechanism of V2O5 on phase transition temperature, there is no proof for it to increase the phase transition temperature. There are many other works also observed with the existence of V2O5, but no one think it will increase the phase transition temperature. For example, the phase transition temperature is intrinsic at about 68when fabricating pure VO2 without any doping. Sometimes there exists V2O5 in pure VO2 film without any doping as well, but no evidence of phase transition temperature increased at all. Therefore, we can’t draw a conclusion that the existence of V2O5 will increase the phase transition temperature of VO2 film.

We tried our best to modify and improve the manuscript according to your comments. And we used the “Track Changes” function in Microsoft Word to mark the revision, so that all the changes can be easily noticed by the editors and reviewers.

We appreciate for your constructive and helpful comments on our manuscript, and hope that the correction and response will meet your requirements.

Round  3

Reviewer 1 Report

This paper may now be published.

Author Response

Dear Reviewer:

We appreciate for your constructive and helpful comments on our manuscript.

Reviewer 4 Report

Dear Colleagues,

I think your work is interesting for the experts.

Still, in your next studies, I would recommend you to pay attention to two problems:

1.     the physical mechanism for reducing the phase transition temperature when doping a VO2 film with tungsten atoms;

2.     influence of V2O5 on the phase transition temperature. You answer (“…but no one think it will increase the phase transition temperature…” and “Therefore, we can’t draw a conclusion that the existence of V2O5 will increase the phase transition temperature of VO2 film”) is insufficient. In order to understand my point of view, I would propose you to discuss it answering the following questions:

1) What is the value of the phase transition temperature of V2O5?

2) What is the value of the phase transition temperature of VO2?

3) What can be the value of the phase transition temperature of a 0.95 × V2O5 + 0.05 × VO2 solid solution?

4) What can be the value of the phase transition temperature of a solid solution of 0.9 × V2O5 + 0.1 × VO2?

5) What can be the value of the phase transition temperature of a solid solution of 0.85 × V2O5 + 0.15 × VO2?

etc.

If, answering these questions, you come to the conclusion that the influence of the V2O5 component on the phase transition is possible, then it is desirable to write about this in the article. I have no doubt that this influence exists.

Author Response

Dear Reviewer,

Thank you very much for carefully review and constructive comments on our manuscript entitled “Tuning the doping ratio and phase transition temperature of VO2 thin film by dual-target co-sputtering” (Manuscript ID: nanomaterials-490208). We studied the comments carefully and tried our best to revise and improve the manuscript according to your comments. The point to point responses to your comments are listed as following:

Point 1: The physical mechanism for reducing the phase transition temperature when doping a VO2 film with tungsten atoms;

Response 1: We have already answered your question in your previous comment as follow: “The reason why doping affects the phase transition temperature of VO2 crystal has not been uniformly explained by theory. According to the existing theories, the phase transition temperature of VO2 film reduced by W doping may be due to the substitution of W6+ to V4+ in d orbital. The introduction of excess electrons reduces the band gap of the d orbit and reduces the phase transition temperature, which can be seen in reference [19]. We added the explanation to the related part of the manuscript.” You can see it in our manuscript line 236- line 240.

We’ve also added the explanation for the physical mechanism in the manuscript as “The reason why doping affects the PTT of VO2 crystal has not been uniformly explained by theory. According to the existing theory, the PTT of VO2 film is reduced by W doping, which may be due to the substitution of W6+ to V4+ in d orbital. The introduction of excess electrons reduces the band gap of the d orbit and reduces the PTT [19].”

Point 2:

Influence of V2O5 on the phase transition temperature. You answer (“…but no one think it will increase the phase transition temperature…” and “Therefore, we can't draw a conclusion that the existence of V2O5 will increase the phase transition temperature of VO2 film”) is insufficient. In order to understand my point of view, I would propose you to discuss it answering the following questions:

1) What is the value of the phase transition temperature of V2O5?

2) What is the value of the phase transition temperature of VO2?

3) What can be the value of the phase transition temperature of a 0.95×V2O5 + 0.05×VO2 solid solution?

4) What can be the value of the phase transition temperature of a solid solution of 0.9×V2O5 + 0.1×VO2?

5) What can be the value of the phase transition temperature of a solid solution of 0.85×V2O5 + 0.15×VO2?

Response 2:

1) The value of the phase transition temperature of V2O5 is 257.[1,2]

2) The value of the phase transition temperature of VO2 is 68.[1,2]

It is not a solid solution of VO2 and V2O5 when both VO2 and V2O5 exist in the film simultaneously. It is only a mixture for them in polycrystalline state. Therefore, the phase transition temperature does not change with the content ratio of VO2 and V2O5 at all. There are two phase transition temperatures from VO2 and V2O5 separately. If the existence of V2O5 will influence the phase transition temperature of the film according to its content, then it should be much higher than the intrinsic phase transition temperature 68℃ of VO2 when without doping. However, the result reported by Ma et al. apparently show that the phase transition temperature of VO2 film with V2O5 is at 67.5,[3] which is almost the same as that of VO2’s. This indicates that V2O5 has no effect on the phase transition temperature of VO2 film at all.

([1] Elizabeth E. Chain. Optical properties of vanadium dioxide and vanadium pentoxide thin films[J]. APPLIED OPTICS, 1991, 30(19): 2782-2787.

[2] Piccirillo C, Binions R, Parkin I P. Synthesis and Functional Properties of Vanadium Oxides: V2O3, VO2, and V2O5 Deposited on Glass by AerosolAssisted CVD[J]. Chemical Vapor Deposition, 2007, 13:145-151.

[3] Ma Hongping, Xu Shiqing. VO2 Thin Films Prepared by V2O5 Melting Formation Thin Films Method[J]. RARE METAL MATERIALS AND ENGINEERING, 2004, 33(3): 317-320.)

We tried our best to modify and improve the manuscript according to your comments. And we used the “Track Changes” function in Microsoft Word to mark the revision, so that all the changes can be easily noticed by the editors and reviewers.

We appreciate for your constructive and helpful comments on our manuscript, and hope that the correction and response will meet your requirements.
